# Animal Models for Gammaherpesvirus Infections: Recent Development in the Analysis of Virus-Induced Pathogenesis

**DOI:** 10.3390/pathogens9020116

**Published:** 2020-02-12

**Authors:** Shigeyoshi Fujiwara, Hiroyuki Nakamura

**Affiliations:** 1Department of Allergy and Clinical Immunology, National Research Institute for Child Health and Development, Tokyo 157-8535, Japan; nakamura-hry@ncchd.go.jp; 2Division of Hematology and Rheumatology, Department of Medicine, Nihon University School of Medicine, Tokyo 173-8610, Japan

**Keywords:** Epstein–Barr virus, Kaposi’s sarcoma-associated herpesvirus, gammaherpesvirus, animal model, humanized mouse, non-human primates, rabbits, lymphoproliferative disease, lymphoma

## Abstract

Epstein–Barr virus (EBV) is involved in the pathogenesis of various lymphomas and carcinomas, whereas Kaposi’s sarcoma-associated herpesvirus (KSHV) participates in the pathogenesis of endothelial sarcoma and lymphomas. EBV and KSHV are responsible for 120,000 and 44,000 annual new cases of cancer, respectively. Despite this clinical importance, no chemotherapies or vaccines have been developed for virus-specific treatment and prevention of these viruses. Humans are the only natural host for both EBV and KSHV, and only a limited species of laboratory animals are susceptible to their experimental infection; this strict host tropism has hampered the development of their animal models and thereby impeded the study of therapeutic and prophylactic strategies. To overcome this difficulty, three main approaches have been used to develop animal models for human gammaherpesvirus infections. The first is experimental infection of laboratory animals with EBV or KSHV. New-world non-human primates (NHPs) and rabbits have been mainly used in this approach. The second is experimental infection of laboratory animals with their own inherent gammaherpesviruses. NHPs and mice have been mainly used here. The third, a recent trend, employs experimental infection of EBV or KSHV or both to immunodeficient mice reconstituted with human immune system components (humanized mice). This review will discuss how these three approaches have been used to reproduce human clinical conditions associated with gammaherpesviruses and to analyze the mechanisms of their pathogenesis.

## 1. Introduction: Strategies to Generate Animal Models for Human Gammaherpesvirus Infections

Approximately 15% of human malignancies are attributed to infection [1], and the two human gammaherpesviruses, Epstein–Barr virus (EBV) [2] and Kaposi’s sarcoma-associated herpesvirus (KSHV) [3], are responsible for 120,000 and 44,000 annual new cases of cancer, respectively [1]. In addition, human gammaherpesviruses, especially EBV, are etiologically involved in various other diseases (Table 1). However, no effective chemotherapy or prophylactic vaccines are available for the two viruses. One reason for this delay in research is the absence of suitable animal models for these viruses. Humans are the only natural host of EBV and KSHV, and very limited species of laboratory animals are susceptible to their experimental infection; this strict host tropism has hampered the development of suitable animal models of human gammaherpesvirus infections. In addition, most human diseases caused by gammaherpesviurses, especially malignancies and autoimmune diseases, are of multifactorial etiology and develop in a small fraction of infected hosts, usually after a long incubation period. Host genetic factors and environmental factors are involved in the pathogenesis of these diseases. This complicated pathogenesis makes modeling of diseases caused by gammaherpesviruses in animal models more challenging. 

Human gammaherpesviruses are classified into the two genera, lymphocryptovirus (LCV) and rhadinovirus (RV), with EBV belonging to the former and KSHV the latter. Beside properties common to all herpesviruses, these two viruses share a number of similar traits, including B-cell tropism and etiological involvement in malignant diseases [2,3]. Properties shared by these viruses allow similar approaches for the development of animal models, and three main approaches have been employed for the purpose (Figure 1, Table 2) [4]. The first is experimental infection of laboratory animals with human gammaherpesviruses. New-world non-human primates (NHPs), such as cotton-top tamarins and common marmosets, and rabbits have been mainly used in this approach [5,6]. The second is exploitation of gammaherpesviruses inherent to laboratory animals. Viruses naturally infecting NHPs and mice have been mainly explored in this approach. Most old-world NHP species and some of new-world NHPs are persistently infected with ubiquitous gammaherpesviruses that are inherent to each species and have a remarkably similar genomic organization and biological properties as those of human gammaherpesviruses [7,8]. These NHP gammaherpesviruses, especially those of rhesus macaques (*Macaca mulatta*), have been extensively explored as models of human EBV and KSHV infections. Mice are also susceptible to persistent infection with a gammaherpesvirus, murine gammaherpesvirus 68 (MHV-68) [9], that targets B cells and causes B-cell lymphoproliferative disease (LPD). The third is humanized mice. Major components of the human immune system can be reconstituted in severely immunodeficient mice following transplantation with human hematopoietic stem cells (HSCs) [10,11,12]. It may be more exact to designate these mice “humanized immune system mice”, since other types of humanized mice, such as “humanized liver mice”, are available now. However, if not otherwise defined, the term “humanized mouse” indicates “humanized immune system mouse” in this article. As human B cells, the major target of both EBV and KSHV, are reconstituted in humanized mice, they can be infected parenterally with both viruses and recapitulate persistent infection, immune responses, and pathogenesis by these viruses [13,14]. This review will discuss how these three approaches have been utilized to recapitulate pathophysiological conditions induced in humans by gammaherpesvirus infections and to analyze the mechanisms of their pathogenesis. In addition to the three approaches described above, immunodeficient mice transplanted with patient-derived tumor cells or with tumor-derived cell lines have been used as in vivo models of gammaherpesvirus-associated tumors. These models have been mainly used to evaluate the effect of experimental therapeutics in vivo and in most cases have not been intended to investigate the pathogenesis of these viruses. These xenograft models are not therefore discussed in this article. Transgenic mouse models for functional analysis of oncoproteins encoded by human gammaherpesviruses have been reviewed recently [15,16] and are not discussed here. 

## 2. Biology and Pathogenesis of EBV and KSHV

### 2.1. EBV

EBV is a ubiquitous virus that persistently infects more than 90% of adult populations in the world [2]. EBV persists for the life of its hosts as a latent infection in memory B cells. EBV is orally transmitted via saliva and most primary EBV infections occur in childhood and present almost no symptoms. However, if primary infection delays until adolescence or later, the virus causes infectious mononucleosis (IM) in 25–74% of infected hosts [51]. The main clinical manifestations of IM include fever, throat pain, lymphadenopathy, liver dysfunction, and CD8+ T-cell lymphocytosis. These manifestations are thought to be mainly generated by excessive T-cell immune responses to EBV-infected cells [52]. EBV has a unique biological activity to transform human B cells and establish lymphoblastoid cell lines. This transformation also occurs in vivo, and transformed lymphoblastoid cells may proliferate autonomously to induce LPD, if immunosurveillance fails [52]. Cellular immunity is thought to play a critical role in protection against both acute and persistent infection with EBV, with CD8+ cytotoxic T cells (CTLs) playing a central role [53]. A number of EBV epitopes restricted by MHC class I and class II have been identified [53]. Innate immunity also plays an important role that is beginning to be elucidated [54]. EBV is a tumor virus originally identified in Burkitt lymphoma (BL) and is etiologically linked to other malignancies such as Hodgkin lymphoma (HL), diffuse large B-cell lymphoma (DLBCL), LPD of immunocompromised hosts, nasopharyngeal carcinoma, and gastric carcinoma [2]. EBV is also implicated in autoimmune diseases, including rheumatoid arthritis (RA) and multiple sclerosis (MS) [2]. For more details of human diseases caused by EBV infection, see Table 1. 

### 2.2. KSHV

The genus rhadinovirus can be further classified into the two groups, rhadinovirus 1 (RV1) and RV2. KSHV, a member of the former, is not as ubiquitous as EBV and its seroprevalence varies in different areas in the world: 0–5% in North and South America, northern Europe, and Japan; around 10% in Mediterranean countries; and 30–50% in sub-Saharan Africa [55]. KSHV is thought to be transmitted mainly via saliva [56], and primary infection in immunocompetent children commonly induces nonspecific symptoms, including fever, rash, and upper respiratory tract symptoms [57]. KSHV has a broader spectrum of cellular targets compared with EBV, and infects B cells, endothelial cells, dendritic cells, monocytes, epithelial cells, and fibroblasts [56]. The site of lifelong KSHV persistence is thought to be B cells, but in contrast with EBV, KSHV does not have the capacity to transform B cells [3]. Both innate and adaptive cellular responses are thought play critical roles in the immune control of KSHV, although information on viral epitopes recognized by KSHV-specific CTLs is limited [58]. KSHV is etiologically linked to the three malignant or lymphoproliferative diseases, Kaposi’s sarcoma (KS), primary effusion lymphoma (PEL), and multicentric Castleman’s disease (MCD), all of which are frequently associated with immunodeficiency caused by HIV (Table 1) [3]. 

## 3. Analysis of EBV-Induced Pathogenesis in Animal Models

### 3.1. Symptomatic Primary EBV Infection or Infectious Mononucleosis 

#### 3.1.1. Experimental Infection of Naïve Rhesus Macaques with Rhesus LCV

The rhesus LCV (rhLCV; *Macacine gammaherpesvirus 4* in International Committee on Taxonomy of Viruses (ICTV)) has a striking similarity with EBV in both genomic organization and biologic properties. Sequence analysis of the rhLCV genome identified a homolog for each known EBV gene, with higher conservation in lytic-cycle genes (49–98% amino acid identity) than in latent-cycle genes (28–60%) [59]. RhLCV has essentially the same life cycle as EBV, including ubiquitous and life-long asymptomatic persistent infection in host populations with intermittent virus release into saliva [60]. B-cell transforming ability and latent infection in memory B cells are also common between EBV and rhLCV. Most captive rhesus macaques in conventional housing (i.e., not tested for and protected from natural gammaherpesvirus infections) turn rhLCV-seropositive within the first year of life, probably via oral transmission, and therefore establishment of rhLCV-free colonies is a critical requirement in performing in vivo research on rhLCV [7]. In a pioneering work of experimental rhLCV infection in rhesus macaques, Moghaddam and others showed that oral inoculation of rhLCV to naïve rhesus macaques induces acute infection with similar manifestations as those in primary EBV infection in humans [25]. Infected macaques displayed atypical lymphocytosis (1–10 weeks post-infection (p.i.)), lymphadenopathy (3–5 weeks p.i.), increase in CD23+ B cells (1–4 weeks p.i.) in the peripheral blood, and splenomegaly [25,26]. A sensitive RT-PCR test detected the rhesus homolog of EBV-encoded small RNA (EBER) in the peripheral blood in 7 days p.i., and DNA PCR detected rhLCV DNA at 2–3 weeks p.i. [25,26]. This acute phase of infection was followed by asymptomatic persistent infection accompanied by stable anti-rhLCV antibody production [25,26]. It is thus evident that rhLCV induces similar virologic and immunopathological events as those in primary EBV infection in humans and provides an excellent model of both acute and chronic EBV infection. It is, however, not clear whether the atypical lymphocytosis in rhLCV-infected rhesus macaques represents excessive virus-specific CTL responses, as has been demonstrated in IM. It is also unknown whether delayed primary rhLCV infection tends to induce exacerbated T-cell responses and severe symptoms, as are seen in primary EBV infection in humans. 

#### 3.1.2. Experimental Infection of Humanized Mice with EBV

Transplantation of human HSCs into mice of highly immunodeficient strains, including NOD/Shi-*scid Il2rg*^null^ (NOG), Balb/c *Rag2*^−/−^
*Il2rg*^−/−^ (BRG), and NOD/LtSz-*scid Il2rg*^−/−^ (NSG), results in reconstitution of human immune system components, including T cells, B cells, NK cells, monocytes/macrophages, and dendritic cells [10,11,12]. NOG mice and NSG mice differ in the nature of *Il2rg* mutation; while NSG mice have a complete null mutation, NOG mice have a truncation of the intracellular signaling domain of IL-2Rγ but retain the extracellular domain intact. Although NOG mice show lower bone marrow CD45+ engraftment following transplantation with human HSCs as compared with NSG mice, the difference in the development of hematopoietic lineages is minimal between the two mouse strains [61]. Since these humanized mice harbor B cells, the major cellular target of both EBV and KSHV, they have been exploited as a small animal model for both viruses [11,13,14,49,62]. In addition, the reconstituted human immune system components mount cellular and humoral immune responses against these viruses, opening the possibility of reproducing their immune-mediated pathogenesis [44,45,62]. 

One of the most remarkable and consistent changes following intravenous inoculation of EBV to humanized mice is rapid and pronounced increase in the number of T cells in the peripheral blood. Similar to IM, most of these increased T cells belong to the CD8+ lineage and include those reacting specifically to EBV epitopes [44,45,46]. EBV-infected humanized mice also exhibit hepatosplenomegaly and elevated human cytokine levels in the peripheral blood [44,45,46]. It is thus evident that some cardinal features of primary EBV infection in humans are reproduced in humanized mice. However, we need more information to understand how far and how exactly humanized mice recapitulate primary EBV infection in humans. For example, antigen specificity of CD8+ T-cell responses in humanized mice has been only partially elucidated, and we do not know whether the pronounced CD8+ T-cell increase following EBV infection is equivalent to the excessive EBV-specific CTL response in IM [44,45,46]. An important deviation in EBV infection of humanized mice from that of humans is the route of transmission. Oral transmission has not been feasible in humanized mice, possibly because their oropharyngeal epithelium has the murine origin. This may have a significant effect on the virologic and immunological consequences of acute EBV infection. 

Investigation in humanized mice suggested that a specific subclass of NK cells plays a critical role in controlling acute EBV infection [47]. In about 4 weeks after EBV infection of humanized mice, a specific increase in the number of NK cells of an early-differentiation phenotype (CD56^dim^ NKG2A^+^ KIR^−^) was demonstrated [47]. Depletion of NK cells with anti-NKp46 antibody in these mice increased the risk of lymphoma development and enhanced CD8+ T-cell responses to the virus [47]. Consistent with this result, a subsequent study in patients with IM demonstrated a critical protective role for NK cells with the same early-differentiation phenotype [63]. It is noteworthy that an observation first made in humanized mice was later confirmed in humans, clearly demonstrating the value of the humanized mouse model. 

In rare cases, primary EBV infection is complicated by hemophagocytic lymphohistiocytosis (HLH), a syndrome caused by overproduction of cytokines by activated T cells and macrophages [64]. Sato and others described manifestations of HLH in EBV-infected humanized mice, including IFN-γ hypercytokinemia and prominent hemophagocytosis in the bone marrow and other organs, suggesting that humanized mice could be used as a model of EBV-associated HLH [46]. This model, however, did not reproduce a unique feature of HLH following primary EBV infection, viz. proliferation of EBV-infected T or NK cells. Imadome and others transplanted peripheral blood mononuclear cells isolated from patients with EBV-HLH to immunodeficient mice and succeeded in reproducing EBV-positive T-cell lymphoproliferation and human hypercytokinemia in mice, demonstrating that EBV-infected T cells are the main source of cytokine overproduction in patients with EBV-HLH [65]. 

The function of reconstituted human immune system in the present forms of humanized mice is suboptimal and requires further improvement, especially when we aim at reproducing immune-mediated pathogenesis. In this context, a number of genetically modified versions of immunodeficient mouse strains have been generated and have brought significant improvement. For example, introduction of the human MHC class I transgene to NSG mice improved CD8+ CTL responses to viruses, including EBV [66]. Knock-in insertion of human genes encoding macrophage colony-stimulating factor (M-CSF), IL-3, granulocyte macrophage colony-stimulating factor (GM-CSF), and thrombopoietin to BRG mice enhanced the development of human myeloid and monocyte/macrophage lineages and improved innate immune responses to viral and bacterial infections [67]. 

#### 3.1.3. Experimental Infection of Mice with MHV-68

The murine gammaherpesvirus 68 (MHV-68, *Murid gammaherpesvirus 4*) shares a number of common properties with KSHV and EBV in life cycle and pathogenesis [9]. MHV-68 persists in memory B cells and induces B-cell LPD in immunocompromised hosts [9]. MHV-68 belongs to the genus rhadinovirus and its genome exhibits extensive homology with KSHV, with more restricted homology to EBV [41]. MHV-68 encodes proteins with sequence similarity with KSHV proteins critically involved in its pathogenesis, including latency-associated nuclear antigen (LANA), the KSHV complement control protein (KCP), a D-type cyclin (v-cyclin), a bcl-2 homolog with anti-apoptotic function (v-bcl-2), and viral G protein-coupled receptor (GPCR) (vGPCR) [41]. Mice infected with MHV-68 exhibit manifestations similar to IM, including CD8+ T-cell lymphocytosis, polyclonal B-cell activation with production of autoantibodies, and splenomegaly [42,68]. The CD8+ T-cell lymphocytosis in MHV-68-infected mice is characterized by a selective expansion of Vβ4+ cells that are not specific to MHV-68 epitopes [42]. In contrast, EBV-specific CD8+ T-cells are mainly expanded in patients with IM, as described above. Interestingly, a peak of viral load in the spleen at 15 days p.i., a characteristic feature of IM-like syndrome in infected adult mice, was not observed in infected neonates [69]. Furthermore, the Vβ4+ CD8+ T-cell expansion was not observed and splenomegaly was less pronounced in infected neonates, suggesting that the development of MHV-68-induced IM-like syndrome is age-dependent, as is that in EBV-induced IM [69].

### 3.2. EBV-Positive Lymphoproliferative Diseases and Lymphomas in Immunocompromised Hosts

As mentioned earlier, in immunocompromised conditions caused by immunosuppressive regimens for transplant recipients, HIV infection, or primary immunodeficiency, proliferation of EBV-transformed B lymphoblastoid cells continues uncontrolled and may lead to the development of EBV-positive LPD or lymphoma [2,52]. Typically, EBV-associated LPD is characterized by polymorphic or immunoblastic histology and exhibits oligoclonal proliferation of EBV-transformed lymphoblastoid cells [70]. However, EBV-positive lymphomas with the histology of BL, HL, or DLBCL also develop in increased frequencies in these immunocompromised conditions, especially in HIV-infected individuals [70].

#### 3.2.1. EBV-Positive LPD in Laboratory Animals Induced by Experimental EBV Infection 

Early models of EBV infection mainly used new-world NHPs as surrogate hosts. Parenteral inoculation of cotton-top tamarins (*Saguinus oedipus*) with EBV resulted in the development of EBV-positive LPD that was later shown to exhibit the latency III type EBV gene expression [5,18,19]. This model was used to evaluate an early experimental EBV vaccine [71]. Experimental infection of common marmosets (*Callithrix jacchus*) with EBV also induced LPD [5,20]. Recently, however, as old-world NHPs infected with their own endogenous LCVs and humanized mice infected with EBV have become prevailing as animal models, these new-world NHPs are less frequently used. 

Rabbits have been also used to recapitulate features of EBV infection in humans [6]. Intravenous inoculation of EBV to New Zealand White rabbits followed by immunosuppression with cyclosporine A induced proliferation of EBV-positive lymphoblasts with the latency III type of viral gene expression in the spleen and the liver, indicating the possibility of using rabbits as a model of EBV-associated LPD [17]. The lineage (B or T) of EBV-infected lymphoblasts was not identified in this study.

#### 3.2.2. LPD/Lymphoma Induced by LCVs Inherent to NHPs in Immunocompromised Conditions

RhLCV has been exploited to analyze the relationship between immunodeficiency and LCV-induced LPD. In order to examine whether depletion of CD4+ T cells by simian/human immunodeficiency virus (SHIV) infection results in the development of LCV-positive LPD, as in HIV-infected individuals, rhLCV-naïve rhesus macaques were first infected with SHIV and then challenged with rhLCV [26]. Unexpectedly, severe reduction of CD4+ T cells (<50 cells/μL) did not result in the development of rhLCV-positive LPD. However, rhLCV DNA in the peripheral blood was detected earlier, reached higher levels, and was cleared later in these CD4+ T-cell-deficient monkeys [26]. These results suggest that immune responses independent of CD4+ T cells, possibly innate immune responses, play a significant role in the protection against rhLCV-induced LPD. Although inoculation of rhLCV did not induce LPD, when two severely immunosuppressed animals (CD4+ cells <50 cells/μL) were transplanted with 1 × 10^8^ rhLCV-transformed autologous B cells, one animal developed an overt LCV-positive monoclonal lymphoma [26]. 

Experimental infection of conventionally housed old-world NHPs with simian immunodeficiency virus (SIV) or SHIV resulted in the development of LPDs harboring LCV, providing a model for AIDS-associated lymphomas. Marshall and others reported that among eighteen rhesus macaques that developed tumors following experimental infection with SIV or SHIV, fourteen had lymphoma of DLBCL histology that harbored rhLCV [27]. These rhesus macaques had probably been infected with rhLCV transmitted naturally from other animals in the same colony before the challenge with SIV/SHIV. 

The development of NHP models of organ transplantation concurrently provided NHP models for EBV-associated post-transplant lymphoproliferative disorder (PTLD). A fraction of old-world NHPs that were treated with immunosuppressive drugs and received allogeneic grafts were found to develop LPDs that harbored their inherent LCVs [28,37,38]. Schmidtko and others reported that among 160 consecutive renal transplant recipients of cynomolgus monkeys (*Macaca fascicularis*), 5.6% developed lymphoproliferative lesions similar to human PTLD in incidence, morphology, and immunophenotype [37]. Lymphoblastoid cells in these lesions were shown to be infected with LCV inherent to cynomolgus macaques (cyLCV; *Macacine gammaherpesvirus 10*) [39,40]. 

#### 3.2.3. LPD/Lymphoma Induced by EBV Infection in Humanized Mice 

EBV infection of humanized mice induces EBV-positive LPD with similar features as those of EBV-associated LPD of immunocompromised hosts, including immunoblastic or DLBCL-like histology, activated B-cell phenotype, and EBV gene expression of the latency III type [13,44,45]. The dose of EBV inoculate is an important factor influencing the outcome of infection; higher doses (>10^2^ 50% transforming dose (TD_50_)) tended to induce LPD, whereas lower doses (<10^1^ TD50) tended to result in persistent infection without any signs of disease [44]. Since depletion of CD4+ or CD8+ T cells resulted in more aggressive and fatal disease, this LPD is considered to be under T-cell immunosurveillance [45,72]. It may be interesting to examine how immunosuppressive drugs affect the incidence and severity of EBV-induced LPD in humanized mice. A modified version of humanized mouse has been recently employed to reproduce EBV-associated LPD. In this model, human cord blood mononuclear cells (depleted for CD34+ HSCs in some cases) were inoculated with EBV in vitro and then transplanted intraperitoneally to immunodeficient mice [73,74]. This modified model is simple in protocol and is supposed to mount more efficient human T-cell immune responses to EBV [75]. 

Investigation of EBV mutants in humanized mice has provided insights into the mechanisms of the virus-induced lymphomagenesis. A BZLF1-deficient EBV mutant was found to be less efficient in generating LPD in humanized mice, whereas a mutant with enhanced BZLF1 expression was more efficient in inducing LPD [76,77]. Current interpretation of these results is that abortive EBV lytic cycle induced by BZLF1 helps to shape an inflammatory microenvironment that supports the virus-induced lymphoproliferation. This hypothesis has been recently supported by a comprehensive next-generation sequencing (NGS) analysis of EBV genomes in various EBV-associated LPD and lymphomas. Deletions in the viral genome predisposing for initiation of abortive lytic cycle have been identified in neoplastic cells of chronic active EBV infection (CAEBV) (see below), extranodal NK/T-cell lymphoma, and DLBCL [78]. These deletions are considered to provide favorable conditions for neoplastic proliferation of EBV-infected cells. An EBNA3B-deficient EBV mutant was found to have an enhanced ability to induce LPD in humanized mice, suggesting a tumor-suppressor function of this gene [79]. Importantly, mutations in the EBNA3B gene have been identified in EBV-positive DLBCL cases among HIV-infected persons, supporting the tumor-suppressing role for the viral protein [79]. 

Recently, Lee and others investigated the effects of lymphocyte profile on the development of EBV-induced lymphoma subtypes in humanized mice and found that a fraction of humanized mice in which T cells were predominantly reconstituted developed lymphomas of HL-like histology with EBV-infected cells of Hodgkin–Reed/Sternberg (HRS)-like morphology [48]. These HRS-like cells displayed the latency II pattern of EBV gene expression and expressed the HL markers, CD15 and CD30. These cells carried hypermutation in the Ig heavy chain variable region, being consistent with the post-germinal center B-cell origin of HL. It may be thus possible to model some aspects of the pathogenesis of EBV-positive HL in humanized mice. Humanized mice with predominant B-cell reconstitution, in contrast, developed lymphomas of DLBCL-like histology [48]. 

Another recent study with humanized mice revealed an essential role for the interaction of exosomes released from EBV-infected B cells and macrophages in EBV-induced lymphoproliferation [80]. This study indicated that exosomes released from EBV-infected B cells deliver microRNAs encoded by the BART region of the viral genome to macrophages and induce their immune regulatory phenotype [80]. This exosome-mediated cellular communication appears to contribute the formation of an inflammatory microenvironment suitable for EBV-induced lymphoproliferation.

#### 3.2.4. LPD Induced by Murine Gammaherpesvirus 68

MHV-68 induces B-cell LPD in immunocompromised hosts. Intranasal inoculation of MHV-68 to BALB/c mice resulted in the development of B-cell LPD in 9–20% of infected hosts after long incubation period (>9 months) [43]. This incidence increased to 60% when cyclosporin A was administered to infected animals [43]. Immunodeficiency due to homozygous null mutations of the IFN-γ receptor gene also facilitated induction of B-cell LPD by MHV-68 [81].

### 3.3. Oral Hairy Leukoplakia

Oral hairy leukoplakia (OHL) is an epithelial lesion caused by focal lytic replication of EBV and found mainly in oral mucosa of HIV-infected individuals [2]. Experimental infection of rhLCV in rhesus macaques that had been infected with SIV resulted in lytic rhLCV infection in epithelial cells and recapitulated OHL-like lesions in the tongue, esophagus, and other sites [29,30]. These lesions exhibit characteristic histology of OHL, such as hyperkeratosis or parakeratosis with demonstration of virions with herpesvirus-like morphology by electron microscope. It is thus considered that EBV’s dual tropism for B cells and epithelial cells is conserved in LCVs inherent to other old-world NHPs. 

### 3.4. EBV-Associated T/NK-Cell Lymphoproliferative Diseases

Beside the major cellular targets of B cells and epithelial cells, T cells and NK cells are occasionally infected with EBV, resulting in T/NK-cell LPDs in rare cases [64]. CAEBV, a prototype of EBV-associated T/NK-cell LPDs, is clinically characterized by prolonged manifestations of IM, including fever, lymphadenopathy, and liver dysfunction [82]. The clinical picture of CAEBV can be viewed from two aspects; one is neoplastic proliferation of EBV-infected T cells or NK cells and the other is multi-organ inflammation induced by overproduction of cytokines by these infected cells [83]. These two aspects of CAEBV have been reproduced in a murine xenograft model [65]. Imadome and others transplanted peripheral blood mononuclear cells isolated from CAEBV patients into immunodeficient mice demonstrated engraftment of the same clone of EBV-infected T/NK cells as that proliferated in the respective patients [65]. High levels of human cytokines, including IL-8, INF-γ, and CCL5 (RANTES), were detected in the peripheral blood of these mice, demonstrating that EBV-infected T/NK cells are mainly responsible for cytokine overproduction in CAEBV [65]. Interestingly, in vivo growth of these EBV-infected T/NK cells was dependent on co-presence of human CD4+ T cells, suggesting that the latter cells provide important signals to facilitate the growth of EBV-infected T and NK cells in vivo and could be a therapeutic target [65]. 

LCVs are mostly B-lymphotropic and cause LPD/lymphoma of B-cell origin, but identification of an exceptional T-cell tropic species of LCV has been reported [84]. A viral DNA fragment was identified in the lesion of cutaneous T-cell lymphoma in a pig-tailed macaque (*Macaca nemestrina*). The same DNA fragment was found also in two CD8+ T-cell lines established from this tumor. Sequencing analysis of this fragment revealed 90% sequence identity with EBV DNA polymerase, and phylogenetic analysis indicated a distinct virus (HV_MNE_) of the LCV genus [84]. Furthermore, herpesvirus-like particles were demonstrated by electron microscopy in the two T-cell lines [84]. Subsequent studies showed that HV_MNE_ transformed rabbit T cells and induced T-cell lymphoma in rabbits [85]. More recent studies suggested that the LCV inherent to the Japanese monkey (*Macaca fuscata*) is also associated with lymphomas of T or NK-cell origin [86,87]. Further research on these viruses might give some hint to the mechanism of EBV infection to T and NK cells and to the enigmatic pathogenesis of EBV-associated T/NK-cell LPDs.

### 3.5. Rheumatoid Arthritis

EBV has been implicated as an environmental factor in the pathogenesis of autoimmune diseases including MS, RA, and systemic lupus erythematosus [88]. However, the absence of a suitable animal model of EBV infection has hampered the direct verification of this hypothesis. In this context, it is noteworthy that EBV infection of humanized mice resulted in erosive arthritis similar to RA [89]. RA is a systemic autoimmune disease with multi-articular erosive arthritis characterized by synovial proliferation and bone destruction [90]. Kuwana and others described the development of erosive arthritis in EBV-infected humanized mice that was characterized by bone destruction, synovial proliferation, and infiltration of human lymphocytes and macrophages in the synovium, all of which represent characteristic histological features of RA [89]. Importantly, the lesion of bone erosion contained human multinuclear osteoclasts and displayed the structure of “pannus”, a histological hallmark of RA [89]. It is thus suggested that cardinal pathological features of RA can be induced and the mechanisms of their pathogenesis can by analyzed in EBV-infected humanized mice. The rheumatoid factor and anticitrullinated protein antibodies, both of which are important features in RA diagnosis, were examined in this study but were not detected. 

### 3.6. Multiple Sclerosis

MS is a systemic autoimmune disease characterized by demyelinization in the central nervous system, and strong epidemiological evidence indicates that EBV infection is almost a prerequisite for its development [91]. Monoclonal antibodies (MAbs) specific to the B-cell marker CD20 were shown to have a pronounced therapeutic effect in MS, indicating a critical role for B cells in its pathogenesis. This clinical effect of anti-CD20 Mab is shared in experimental autoimmune encephalitis (EAE) in common marmosets, a preclinical model of MS. Recently, anti-CD20 MAbs were shown to exert their therapeutic effects through depletion of a specific fraction of B cells that was infected with CalHV-3 (*Callitrichine gammaherpesvirus 3*) [22], a marmoset LCV closely related to EBV [23]. Moreover, infusion of autologous CalHV-3-infected B cells pre-pulsed with the EAE-inducing MOG_34–56_ peptide into common marmosets induced histological signs of meningeal inflammation and immunological features reminiscent of EAE [23]. These findings suggest a critical role for CalHV-3 in EAE pathogenesis [23,24]. Subsequent work demonstrated that CalHV-3-infected B cells have increased capacity of cross-presenting MOG_34–56_ to a specific type of CTLs with the surface phenotype of CD8 + CD56 + CD28-. This particular subset of CTLs had been known to play a critical role in the progression phase of marmoset EAE [24,92]. In CalHV-3-infected B cells, an arginine residue of MOG_34–56_ is citrullinated, and this modification protects the peptide from cleavage by cathepsin G and facilitates its presentation to CD8 + CD56 + CD28- T cells [92]. Activation of autophagy by CalHV-3 infection may also have a role in the enhancement of cross-presentation by the virus [92]. These findings imply that EBV, the human equivalent of CalHV-3, sharing a number of common properties with CalHV-3, including B-cell transformation, plays a similar role in MS pathogenesis. Interestingly, MHV-68 infection has been shown to exacerbate manifestations of the mouse version of EAE induced by MOG_35–55_, suggesting the involvement of gammaherpesviruses in the pathogenesis of MS models of both primates and rodents [93,94]. 

## 4. Analysis of KSHV-Induced Pathogenesis in Animal Models 

### 4.1. Kaposi’s Sarcoma

KS is a vascular endothelial tumor characterized by proliferation of spindle-shaped tumor cells infected with KSHV and prominent neoangiogenesis [95]. KSHV is present in tumor cells of all four epidemiological forms of KS, namely classic, endemic, AIDS-related, and iatrogenic [95].

#### 4.1.1. KS-Like Tumor Induced by Experimental Infection of NHPs with KSHV

Chang and others showed that common marmosets are susceptible to experimental infection with KSHV [21]. Inoculation of the virus to common marmosets resulted in vigorous antibody response, viral DNA load, and LANA expression in peripheral blood and various organs. Importantly, one of these animals that were orally inoculated developed a KS-like skin lesion with KSHV-positive spindle-shaped cell proliferation and leukocyte infiltration [21]. In contrast, experimental transmission of KSHV to rhesus macaques that had been infected with SIV resulted in only low-level viral DNA load in the peripheral blood and did not reproduce any disease conditions [96].

#### 4.1.2. KS-Like Tumor Caused by the Retroperitoneal Fibromatosis-Associated Herpesvirus (RFHV)

Retroperitoneal fibromatosis (RF) is a vascular fibroproliferative disease originally described in captive macaques housed in primate research facilities [97]. Histologically, RF shares some common features with KS, including proliferation of characteristic spindle-shaped cells, infiltration of inflammatory cells, and neoangiogenesis, although its primary localization is retroperitoneum and not the skin. Another similarity between RF and KS is that both tumors develop in host animals suffering lentivirus-induced immunodeficiency [97]. Degenerative PCR identified partial genomic DNA of an RV that was later termed retroperitoneal fibromatosis-associated herpesvirus (RFHV; *Macacine gammaherpesvirus 8*) [35]. A nuclear protein encoded by the LANA homolog of RFHV was found in spindle-shaped cells of RF, and RFHV is now considered as the etiologic agent of RF. The complete genomic sequence of RFHV has been determined and found to be colinear with that of KSHV and has a homolog for every KSHV gene except for ORF11, K5, and K6 [98]. Both KSHV and RFHV belong to the RV1 group. It was recently reported that experimental infection of a conventionally housed rhesus macaque with SIV or SHIV induced a KS-like fibrosarcoma in the colon-rectum region with spindle-shaped cell proliferation [27]. This tumor was shown to be infected with RFHV and to express the RFHV homolog of KSHV LANA [27]. RF is thus a potentially excellent animal model of KS, however, it has not been possible so far to isolate and propagate RFHV in culture, hampering experimental infection of macaques with the virus. Nevertheless, a most recent investigation identified high-level viral DNA in the saliva of RFHV-infected macaques and demonstrated that inoculation of the saliva containing RFHV to naïve macaques resulted in successful infection [36].

#### 4.1.3. KS-Like Tumor Induced by Experimental Infection of the Rhesus Macaque Rhadinovirus, RRV

The rhesus macaque rhadinovirus (RRV, *Macacine gammaherpesvirus 5*) belongs to the RV2 group and is ubiquitous in rhesus macaque populations [31]. The genomic sequence of a strain (17577) of RRV has been obtained and showed extensive homology with KSHV, although it is less homologous to KSHV than RFHV is [32]. RRV has been isolated and can be propagated in culture, facilitating its use in experimental infection in animal models. Co-infection of rhesus macaque with RRV and SIV induced a KS-like lesion attached to the stomach with proliferation of RRV-infected spindle-shaped cells, suggesting that RRV can also induce a KS-like lesion [33]. RF lesions induced either by RFHV or by RRV develop in peritoneal cavity, making a strong contrast with the predominant localization of KS in the skin. 

### 4.2. Primary Effusion Lymphoma

PEL is a B-cell lymphoma found mostly in HIV-infected individuals and localized primarily in pericardial, pleural, or peritoneal cavity [56]. PEL exhibits morphological features of immunoblastic or anaplastic large-cell lymphoma with a surface phenotype of plasma cell differentiation [56]. KSHV is found in tumor cells of all cases of PEL, whereas EBV is found in 70–90% of cases [99,100,101]. It is generally considered that KSHV is largely responsible for the pathogenesis of PEL, whereas the role of EBV is unknown. Recently, McHugh and others reported on interesting interaction of EBV and KSHV in humanized mice [50]. While inoculation of KSHV alone did not result in efficient establishment of persistent infection in humanized mice, co-infection with EBV increased the probability of establishing KSHV persistence [50]. These results are consistent with the previous in vitro finding that co-infection of EBV enhanced the persistence of KSHV in PEL-derived cell lines [102]. This study also showed that co-infection with KSHV enhanced EBV-induced lymphomagenesis in humanized mice [50]. Because co-infection of EBV with knocked-out BZLF gene did not enhance KSHV persistence, initiation of early EBV lytic cycle is thought to play a critical role. EBV/KSHV dually infected mice developed lymphoma in the spleen and peritoneal cavity that harbored both viruses, and cell lines established from this lymphoma exhibited the transcriptional hallmark of PEL and signs of plasma cell differentiation [50]. These results imply that EBV can be positively involved in the process of PEL pathogenesis along with KSHV.

### 4.3. Multicentric Castleman’s Disease

MCD is a lymphoproliferative disease of IgM λ-restricted plasmablasts [3] and characterized by prominent inflammatory symptoms. Experimental co-infection of naïve rhesus macaques with the RRV strain 17577 (RRV_17577_) and SIV induced profound and persistent multicentric angiofollicular lymphadenopathy resembling the multicentric plasma cell form of MCD [34]. In addition, hypergammaglobulinemia and immune-mediated hemolytic anemia were observed in these animals, demonstrating further similarity with MCD [34]. A follow-up study showed that RRV_17577_/SIV co-infection of rhesus macaques can also induce monoclonal non-Hodgkin B-cell lymphoma, as well as RF-like mesenchymal lesion that has been mentioned above [33]. These studies also showed that viral homolog of IL-6 encoded by RRV_17577_ is expressed in the MCD-like LPD and non-Hodgkin lymphoma, suggesting a similar involvement of viral IL-6 homologs in KSHV- and RRV-associated malignancies [33]. Curiously, no similar pathological conditions have been observed in rhesus macaques infected with the other strain of RRV, RRV_H26–95_, suggesting a strain-specific pathogenicity [103].

## 5. Perspective

As mentioned earlier, human gammaherpesviruses are usually carried as asymptomatic persistent infection. Most EBV- and KSHV-associated diseases, especially malignancy and autoimmunity, are likely multifactorial and develop in a small fraction of infected hosts who are under the influence of particular host genetic factors and/or environmental factors. This complicated pathogenesis elevates the hurdle of modeling diseases caused by these viruses, and their exact recapitulation in animals may be difficult. In spite of this difficulty, insights into essential principles in the pathogenesis of gammaherpesviruses have been obtained in animal models and subsequently proven relevant in humans, as described above (e.g., tumor enhancing roles for the abortive EBV lytic cycle, the tumor suppressor function of EBNA3B, and the critical protective role for NK cells of the early differentiation phenotype in primary EBV infection).

Animal models described above may be used to analyze the effect of host genetic factors in the pathogenesis of these viruses. In this context, experiments with MHV-68 may give insights into crucial host genetic factors in gammaherpesvirus-associated diseases. Accumulated knowledge of mouse genetics and advanced techniques in genetic modification have made it possible to introduce almost any desired mutations in the mouse genome and to examine their effects on the outcome of MHV-68 infection. For example, MHV-68 infection of mice knocked-out for genes involved in innate immunity revealed that defects in genes responsible for cytosolic DNA sensing enhances the establishment of MHV-68 latent infection [104]. Similarly, as described already, homozygous null mutations of the IFN-γ receptor gene enhanced the generation of B-cell LPD by MHV-68 [81]. Research in this direction may be easily extended to mouse homologs of human genes that are expected to be involved in immune control of gammaherpesviruses. More recently, application of gene editing techniques has been extended to NHPs. A common marmoset model of severe combined immunodeficiency has been generated by using zinc-finger nucleases and transcription activator-like effector nucleases [105]. Targeted gene modification has become feasible also in macaques [106]. Results obtained from experimental infection of these gene-modified NHPs with gammaherpesviruses should give more direct insights into the role for host genetic factors in human gammaherpesvirus-induced diseases. 

## Figures and Tables

**Figure 1 pathogens-09-00116-f001:**
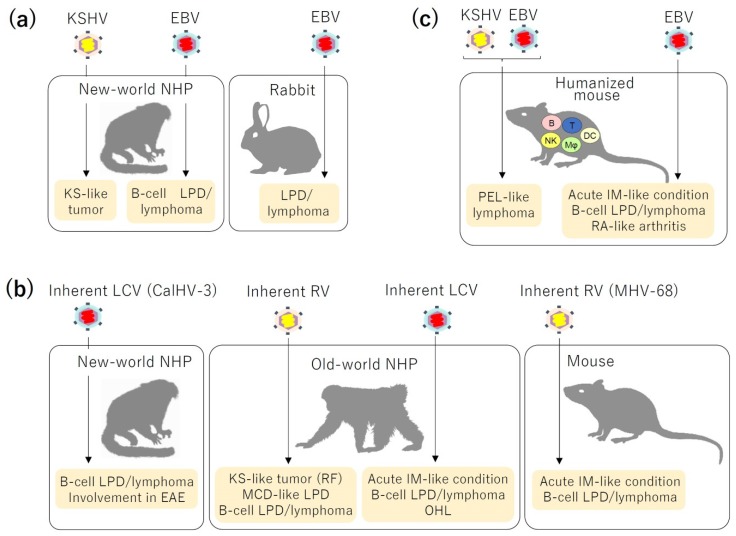
Representative animal models for gammaherpesvirus-induced diseases. Three main approaches to generate animal models for human gammaherpesvirus infections are illustrated, i.e., experimental infection of laboratory animals (new-world NHPs and rabbits) with EBV or KSHV (**a**), infection of laboratory animals with their own inherent LCVs (CalHV-3 for common marmosets, rhLCV for rhesus macaques, and cyLCV for cynomolgus macaques) or RVs (RRV and RFHV for macaques, and MHV-68 for mice) (**b**), and infection of humanized mice with EBV or KSHV or both (**c**). Clinical conditions reproduced in each model are shown below arrows. NHPs, non-human primates; EBV, Epstein–Barr virus; KSHV, Kaposi’s sarcoma-associated herpesvirus; LCVs, lymphocryptoviruses; RVs, rhadinoviruses; CalHV-3, callitrichine gammaherpesvirus 3; rhLCV, rhesus lymphocryptovirus; cyLCV, cynomolgus LCV; RRV, rhesus macaque rhadinovirus; RFHV, retroperitoneal fibromatosis-associated herpesvirus; MHV-68, murine gammaherpesvirus 68; KS, Kaposi’s sarcoma; LPD, lymphoproliferative disease; PEL, primary effusion lymphoma; IM, infectious mononucleosis; RA, rheumatoid arthritis; EAE, experimental autoimmune encephalitis; RF, retroperitoneal fibromatosis; MCD, multicentric Castleman’s disease; OHL, oral hairy leukoplakia.

**Table 1 pathogens-09-00116-t001:** Human gammaherpesvirus-associated diseases.

Virus	Disease Category	Disease
EBV	Acute disease following primary infection	Infectious mononucleosis (IM), EBV-associated hemophagocytic lymphohistiocytosis (EBV-HLH) ^a^
EBV	Opportunistic infection	Posttransplant lymphoproliferative disease (PTLD), AIDS-associated lymphomas, oral hairy leukoplakia (OHL), EBV-positive LPD/lymphoma in primary immunodeficiency
EBV	T/NK-cell lymphoproliferative disease	Systemic chronic active EBV infection (CAEBV), hydroa vacciniforme-like lymphoproliferative disorder, severe mosquito bite allergy
EBV	Malignancy ^b^	Burkitt lymphoma (BL), Hodgkin lymphoma (HL), Diffuse Large B-cell lymphoma (DLBCL), extranodal NK/T-cell lymphoma-nasal type, Aggressive NK-cell leukemia, gastric carcinoma, nasopharyngeal carcinoma, salivary gland carcinoma
EBV	Autoimmune disease ^c^	Rheumatoid arthritis (RA), systemic lupus erythematosus, multiple sclerosis (MS)
KSHV	Malignancy	Kaposi’s sarcoma (KS), Primary effusion lymphoma (PEL)
KSHV	Lymphoproliferative disease	Multicentric Castleman’s disease (MCD)

^a^ EBV-HLH is characterized by proliferation of EBV-infected T cells (or occasionally NK cells). ^b^ In addition to the malignancies listed in this table, EBV infection in malignant cells has been also documented in rare tumors, such as thymoma, systemic EBV-positive T-cell lymphoma of childhood, plasmablastic lymphoma, primary effusion lymphoma, mucocutaneous ulcer, lymphomatoid granulomatosis, leiomyosarcoma, and thymic carcinoma. ^c^ EBV is also implicated in other autoimmune diseases, including Sjögren syndrome, autoimmune thyroiditis, inflammatory bowel disease, type 1 diabetes, scleroderma, myasthenia gravis, and autoimmune hepatitis.

**Table 2 pathogens-09-00116-t002:** Representative animal models of human gammaherpesvirus infections ^a^.

Animal Species	Virus Species	Features of Viral Infection Reproduced	Reference
Rabbit	EBV	LPD/lymphoma, peripheral blood EBV DNA, EBV-specific antibodies	[6,17]
Cotton-top tamarin	EBV	B-cell LPD/lymphoma	[5,18,19]
Common marmoset	EBV	B-cell LPD/lymphoma	[5,20]
	KSHV	KS-like skin tumor	[21]
	Marmoset LCV, *Callitrichine gammaherpesvirus 3* (CalHV-3)	B-cell LPD/lymphoma, implicated in the pathogenesis of EAE (an experimental model of MS)	[22,23,24]
Rhesus macaque	Rhesus LCV (rhLCV), *Macacine gammaherpesvirus 4*	Acute IM-like syndrome, B-cell LPD/lymphoma, OHL, persistent infection	[7,25,26,27,28,29,30]
	Rhesus rhadinovirus (RRV), *Macacine gammaherpesvirus 5*	MCD-like B-cell LPD, B-cell lymphoma, KS-like tumor, persistent infection	[31,32,33,34]
Macaques (rhesus, cynomolgus, pig-tailed)	Retroperitoneal fibromatosis-associated herpesvirus (RFHV), *Macacine gammaherpesvirus 8*	Retroperitoneal fibromatosis (a KS-like tumor)	[27,35,36]
Cynomolgus macaque	Cynomolgus LCV (cyLCV), *Macacine gammaherpesvirus 10*	B-cell LPD/lymphoma	[37,38,39,40]
Mouse	Murine gammaherpesvirus 68 (MHV-68), *Murid gammaherpesvirus 4*	Acute IM-like syndrome, persistent infection, B-cell LPD/lymphoma	[9,41,42,43]
NOG, BRG, NSG mice engrafted with human HSCs	EBV	Acute IM-like syndrome, B-cell LPD/lymphoma, Hodgkin-like lymphoma, EBV-HLH, RA-like arthritis, persistent infection, innate and adaptive immune responses	[11,44,45,46,47,48]
BLT-NSG mice	KSHV	Infection to B cells and macrophages	[49]
NSG mice engrafted with human HSCs	KSHV and EBV	B-cell lymphoma with PEL-like phenotype and gene expression	[50]

^a^ Xenograft models and transgenic mouse models are not included in this table. HSCs, hematopoietic stem cells.

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
