# Peer review of "Animal Models for Gammaherpesvirus Infections: Recent Development in the Analysis of Virus-Induced Pathogenesis"

_pathogens, 2020, doi:10.3390/pathogens9020116_

Round 1
Reviewer 1 Report
This manuscript is an authoritative review of animal models for gammaherpesviruses. It is very well-written and would be an important and well-received review of the cutting edge efforts to model pathogenesis of human gamma-herpesviruses in animals. One of the strengths of the review is the authors’ ability to state very clear conclusions from a large and ever-growing virology literature. The authors are succinct yet forceful.
However, in some specific examples, the succinctness leads to overstatement of controversial areas of gamma-herpes research. These examples should be addressed by modifying the manuscript using a few alternative strategies:
the authors should state that there is controversy, the authors should add citations to the direct, basic research paper, rather than other review papers, for the controversial statements, or the authors should remove the controversial statement.
Those specific areas of controversy are the following:
The percentage of PELs that are co-infected by KSHV and EBV. The authors state that 90% of PELs are co-infected (page 11, line 460). This is a very high number offered without citation to a direct research paper. In fact, the number of co-infections is high, but the exact percentage is stated differently in different articles in the literature. Moreover, it is more accurate to state that the co-infection numbers come from clonally-selected PEL “cell lines,” and not actually primary histologic samples. Therefore, this is overstated and potentially misleading as currently worded. While the research that is reviewed in the ensuing paragraph is important and well-described, the final sentence of the paragraph is also overstated (p. 12, line 472). “….EBV is positively involved in …. PEL…” should be re-stated as “….EBV ‘can’ be positively involved…” or some other less forceful description. Endothelial cells are a lifelong reservoir of KSHV infection (p. 5, line 129). While endothelial cells certainly can be latently infected by KSHV, many virologists don’t believe that endothelial cells are a life-long site of latency. To my knowledge, there is no proof of “life-long latency” in endothelial cells in the literature. If the authors can cite that statement from primary, research papers, they should add it. Otherwise, it is more accurate to state that B cells are the site of life-long latency.
Other concerns:
Many abbreviations are not defined at the position at which they are first used. This could be clarified by adding text to the legend to Figure 1 that directs the reader to “see Tables 1 and 2 and the text for definitions of the abbreviations.” The comparisons of the models to human diseases are very cogent and interesting. However, table 2 should precede Table 1. The actual human diseases should be described first, after which the models for those diseases should follow. Page 5, Line 129, “cites” should be changed to “sites” Paragraph 3.1.2, the authors make the distinction that both NOG and NSG mice have been used to generate humanized immune systems. Perhaps a line that explains the distinction between the genetic background of NOG and NSG would be enlightening. Were there any important differences in using those different backgrounds in modeling herpes infection? The conclusion of the final sentence of paragraph 3.5 is not clear, particularly regarding the rheumatoid factor and anti-citrullinated antibody. Were they not shown, not observed, or not elevated?Author Response
Comment:
This manuscript is an authoritative review of animal models for gammaherpesviruses. It is very well-written and would be an important and well-received review of the cutting edge efforts to model pathogenesis of human gamma-herpesviruses in animals. One of the strengths of the review is the authors’ ability to state very clear conclusions from a large and ever-growing virology literature. The authors are succinct yet forceful.
However, in some specific examples, the succinctness leads to overstatement of controversial areas of gamma-herpes research. These examples should be addressed by modifying the manuscript using a few alternative strategies: the authors should state that there is controversy, the authors should add citations to the direct, basic research paper, rather than other review papers, for the controversial statements, or the authors should remove the controversial statement.
Reply:
We appreciate your favorable comments and constructive criticism. We revised the manuscript in accordance with your review.
Comment:
Those specific areas of controversy are the following:
The percentage of PELs that are co-infected by KSHV and EBV. The authors state that 90% of PELs are co-infected (page 11, line 460). This is a very high number offered without citation to a direct research paper. In fact, the number of co-infections is high, but the exact percentage is stated differently in different articles in the literature. Moreover, it is more accurate to state that the co-infection numbers come from clonally-selected PEL “cell lines,” and not actually primary histologic samples. Therefore, this is overstated and potentially misleading as currently worded.
Reply:
We revised the sentence as follows and cited three additional references (line 482 of the revised manuscript); “..., whereas EBV is found in 70-90% of cases”. The line number corresponds to what is shown when track changes are displayed.
Comment:
While the research that is reviewed in the ensuing paragraph is important and well-described, the final sentence of the paragraph is also overstated (p. 12, line 472). “….EBV is positively involved in …. PEL…” should be re-stated as “….EBV ‘can’ be positively involved…” or some other less forceful description.
Reply:
We modified the sentence as suggested (line 494); “These results imply that EBV can be positively involved in the process of PEL pathogenesis along with KSHV.”
Comment:
Endothelial cells are a lifelong reservoir of KSHV infection (p. 5, line 129). While endothelial cells certainly can be latently infected by KSHV, many virologists don’t believe that endothelial cells are a life-long site of latency. To my knowledge, there is no proof of “life-long latency” in endothelial cells in the literature. If the authors can cite that statement from primary, research papers, they should add it. Otherwise, it is more accurate to state that B cells are the site of life-long latency.
Reply:
We agree with the comment and modified the sentence (line 144); “The site of lifelong KSHV persistence is thought to be B cells, but in contrast...”
Comment:
Many abbreviations are not defined at the position at which they are first used. This could be clarified by adding text to the legend to Figure 1 that directs the reader to “see Tables 1 and 2 and the text for definitions of the abbreviations.”
Reply:
Thank you for the helpful comment and we put the suggested sentence in the legend to Figure 1 (line 56-57).
Comment:
The comparisons of the models to human diseases are very cogent and interesting. However, table 2 should precede Table 1. The actual human diseases should be described first, after which the models for those diseases should follow.
Reply:
We exchanged the relative position of Tables 1 and 2. To do this, we added a sentence in the introduction section (line 37-39); “In addition, human gammaherpesviruses, especially EBV, are etiologically involved in various other diseases (Table 1).
Comment:
Page 5, Line 129, “cites” should be changed to “sites”
Reply:
We corrected the word accordingly (line 144).
Comment:
Paragraph 3.1.2, the authors make the distinction that both NOG and NSG mice have been used to generate humanized immune systems. Perhaps a line that explains the distinction between the genetic background of NOG and NSG would be enlightening. Were there any important differences in using those different backgrounds in modeling herpes infection?
Reply:
We added the following sentence (line 183-188); “NOG mice and NSG mice differ in the nature of Il2rg mutation; while NSG mice have a complete null mutation, NOG mice have a truncation of the intracellular signaling domain of IL-2Rγ but retain the extracellular domain intact. Although NOG mice show lower bone marrow CD45+ engraftment following transplantation with human HSCs as compared with NSG mice, the difference in the development of hematopoietic lineages is minimal between the two mouse strains.”
Comment:
The conclusion of the final sentence of paragraph 3.5 is not clear, particularly regarding the rheumatoid factor and anti-citrullinated antibody. Were they not shown, not observed, or not elevated?
Reply:
The rheumatoid factor and anti-citrullinated antibodies were examined but were not detected in the study. We corrected a spelling error (by → be) in the last sentence and revised the last part of the paragraph (line 404-408); “It is thus suggested that cardinal pathological features of RA can be induced and the mechanisms of their pathogenesis can be analyzed in EBV-infected humanized mice. The rheumatoid factor and anti-citrullinated protein antibodies, both of which are important features in RA diagnosis, were examined in this study but were not detected.”
Reviewer 2 Report
This is an interesting and well written review article about animal models of gammaherpesvirus infection and related human diseases. There are a few minor errors which need to be corrected:
1. At 2.2. KSHV (Line 129 of the manuscript), "The cites of lifelong KSHV..." should be "The sites of lifelong KSHV...".
2. At 3.1.2. Experimental infection of humanized mice with EBV (Line 215 of the manuscript), "...to viral an bacterial infections" should be "...to viral and bacterial infections".
Author Response
Comment:
This is an interesting and well written review article about animal models of gammaherpesvirus infection and related human diseases. There are a few minor errors which need to be corrected:
Reply:
We appreciate your favorable and helpful comments and corrected the spelling errors.
Comment:
At 2.2. KSHV (Line 129 of the manuscript), "The cites of lifelong KSHV..." should be "The sites of lifelong KSHV...".Reply:
We corrected the word accordingly (line 144 of the revised manuscript). The line number corresponds to what is shown when track changes are displayed.
Comment:
At 3.1.2. Experimental infection of humanized mice with EBV (Line 215 of the manuscript), "...to viral an bacterial infections" should be "...to viral and bacterial infections".Reply:
We corrected the word accordingly (line 235).
Reviewer 3 Report
Fujiwara and Nakamura have provided a very well written review paper to cover animal models to study gamma herpesviruses.
One possible downside of the manuscript that I could think about is that the manuscript does not cover some major work which is done in the field. Having said that, I highly appreciate that it is not an easy task to include all of the published work in such a diverse and fast-growing area into a review paper.
All in all, I think that manuscript reads well and can be very informative for MDPI audience.
Author Response
Comment:
Fujiwara and Nakamura have provided a very well written review paper to cover animal models to study gamma herpesviruses.
One possible downside of the manuscript that I could think about is that the manuscript does not cover some major work which is done in the field. Having said that, I highly appreciate that it is not an easy task to include all of the published work in such a diverse and fast-growing area into a review paper.
All in all, I think that manuscript reads well and can be very informative for MDPI audience.
Reply:
We appreciate the favorable comment. We agree that this review may not have covered all major works in the field, although we did our best to do so.
Round 2
Reviewer 1 Report
The authors have been very responsive to reviews.